# Comparative efficacy of robot-assisted therapy associated with other different interventions on upper limb rehabilitation after stroke: A protocol for a network meta-analysis

Qian Liu[1], Zuoyan Liu[2], Yang Xu[2], Li Liu[1], Fang Wang[1], Fanyu Zhao[1], Hong Cheng[3], Xiuying Hu[1] *

1 Innovation Center of Nursing Research, Nursing Key Laboratory of Sichuan Province, West China Hospital, Sichuan University/West China School of Nursing, Sichuan University, Chengdu, China, 2 Department of Rehabilitation Medicine Center, West China Hospital, Sichuan University, Chengdu, China, 3 School of Automation Engineering, University of Electronic Science and Technology of China, Chengdu, China

* huxiuying@scu.edu.cn

**Data Availability Statement:** No datasets were generated or analysed during the current study. All

## Abstract

### Introduction

Post-stroke movement disorders are common, especially upper limb dysfunction, which seriously affects the physical and mental health of stroke patients. With the continuous development of intelligent technology, robot-assisted therapy has become a research hot-spot in the upper limb rehabilitation of stroke patients in recent years. Many scholars have also integrated robot-assisted therapy with other interventions to improve rehabilitation outcomes. However, there is a lack of research to determine which auxiliary intervention is the best. Therefore, this protocol aims to guide the development of a network meta-analysis, which helps determine the most suitable auxiliary interventions for robot-assisted therapy.

### Methods and analysis

Published randomized controlled trials will be included if robot-assisted therapy or robot-assisted therapy associated with other different interventions was applied in stroke patients with upper limb dysfunction in the experimental group and usual rehabilitation treatment and care was applied in the control group. CINAHL, PubMed, Web of Science, MEDLINE, Embase, CNKI, and Wanfang electronic databases will be searched. Studies should be published between January 1, 2013, and December 31, 2023. Two reviewers will independently select studies and extra data, and assess the quality of the included studies. The risk of bias will be evaluated based on the Cochrane Collaboration's risk of bias tool. The evidence quality will be measured according to the Grading of Recommendations Assessment, Development and Evaluation. A network meta-analysis will be conducted by using STATA version 15.0 and R version 4.1.3. The probabilities of rehabilitation interventions will be ranked according to the surface under the cumulative ranking curve.

relevant data from this study will be made available upon study completion.

**Funding:** This study is funded by the project from the West China Hospital of Sichuan University (Grant No. HXDZ21003).

**Competing interests:** The authors have declared that no competing interests exist.

## Ethics and dissemination

Ethical approval is not needed for reviewing published studies. The results will be submitted to a journal.

## Trial registration

**PROSPERO registration number:** CRD42023486570.

## Introduction

With the deepening of global aging, the incidence rate of stroke has shown a high growth trend in recent years. The disability rate of stroke is also high, and it has become a major public health problem faced by countries around the world. Stroke is the second leading cause of early death and secondary disability, especially upper limb dysfunction [1]. Approximately 80% of patients have upper limb dysfunction after stroke, and 50% of patients continue to experience upper limb dysfunction 4 years after stroke [2]. Upper limb motor dysfunction can lead to a decline in self-care ability and reduce the level of activities of daily living, hindering their normal return to society after discharge and seriously affecting their physical and mental health. Therefore, an increasing number of scholars have paid attention to the improvement of upper limb function in patients with stroke.

Rehabilitation training is the most effective way to reduce the disability rate of stroke, but there is a serious shortage of rehabilitation therapists to provide rehabilitation training for patients worldwide [3]. With the continuous development of intelligent technology, robot-assisted therapy has become a research hotspot in the rehabilitation of upper limb function in stroke patients in recent years. Robot-assisted therapy can enhance the intensity of rehabilitation training for stroke patients, enrich the rehabilitation process, and reduce the workload of rehabilitation therapists. However, the results of previous studies are not consistent. A meta-analysis has suggested that robot-assisted therapy can significantly improve the upper limb motor function of stroke patients compared with conventional rehabilitation therapy [4]. In contrast, a multicenter randomized controlled trial has shown that robot-assisted therapy can not significantly improve upper limb function in stroke patients with moderate or severe upper limb dysfunction [2]. Therefore, it is necessary and essential to combine robot-assisted therapy with other rehabilitation programs to improve the upper limb motor function of stroke patients. To date, some scholars have integrated robot-assisted therapy with noninvasive brain stimulation, electromyography, virtual reality technology, and brain computer interfaces to enhance the rehabilitation outcomes of stroke patients [5–7]. Giacobbe et al. [8] demonstrated that transcranial direct current stimulation can improve the effects of rehabilitation training when applied immediately prior to robot-assisted therapy. Robots based on electromyography can detect residual electromyographic signals of affected limbs in real- time and integrate autonomous movement intentions into training, aiming to maximize the voluntary effort of post-stroke training [9, 10]. Virtual reality technology can increase multisensory feedback, stimulate the reorganization of the cerebral cortex, and activate the contralesional primary sensorimotor cortex [11, 12]. Combined with robot-assisted therapy, it can more effectively improve upper limb function in stroke patients [13]. In addition, the combination of robot-assisted therapy and brain computer interfaces can promote the reorganization of brain functional connections [14], and significantly improve the rehabilitation outcomes of

upper limb function in stroke patients [15]. In total, the combination of robot-assisted therapy and other different rehabilitation programs can improve the effectiveness of rehabilitation training. However, there is a lack of research to compare which rehabilitation program combined with robot-assisted therapy has the best effect, and the evidence needs to be updated in real time. As a consequence, to provide important evidence for the rehabilitation and nursing decisions of stroke patients with upper limb dysfunction, this protocol aims to guide a network meta-analysis that can evaluate the effectiveness of robot-assisted therapy combined with other different interventions and rank them.

## Methods

In accordance with the Preferred Reporting Items for Systematic Review and Meta-Analysis Protocols [16], we developed this protocol for a network meta-analysis, which was registered on the PROSPERO platform (CRD42023486570).

### Eligibility criteria

**Types of participants.**   The inclusion criteria will be: stroke patients aged 18 years old and above; with upper limb dysfunction caused by stroke; first six months after stroke. Stroke patients with malignant tumors, organ failure, and other neurological diseases that may cause motor deficits will be excluded.

**Types of interventions.**   Trials will be included if the rehabilitation training is robot-assisted and aimed at improving upper limb function. Trials that added other different interventions, such as transcranial direct current stimulation, transcranial alternating current stimulation, transcranial random noise stimulation, transcranial magnetic stimulation, brain-computer interface, virtual reality, noninvasive brain stimulation, electromyographic feedback and internet-based remote rehabilitation, to improve upper limb function will be included. The dosage and intensity of these interventions will be limited to once a day and at least three times a week. Multicomponent interventions will be excluded.

**Types of controls.**   The control group received the same amount of non-robotic rehabilitation training, such as usual rehabilitation treatment and care, conventional treatment, motion exercise, occupational therapy, and physical therapy.

**Types of outcomes.**   The primary outcomes will focus on upper limb function, whereas secondary outcomes will be activities of daily living. Upper limb function may be measured by the Fugl-Meyer Assessment for Upper Extremity, Ashworth Scale, modified Ashworth Scale, Wolf Motor Function Test, the Box and Block Test, Action Research Arm Test, or Motor Activity Log. Activities of daily living may be evaluated by the Barthel Index score, modified Barthel Index, the Functional Independence Measure, or the Community Integration Questionnaire.

**Types of studies.**   Randomized controlled trials written in English or Chinese will be included.

### Data sources and search strategy

The professional search will combine the Medical Subject Headings with free words. The search items will include: stroke*, apoplexy*, cerebral vascular accident, brain vascular accident, cerebral vascular disorders, brain vascular disorders, intracranial hemorrhage, cerebral hemorrhage, brain hemorrhage, subarachnoid hemorrhage, cerebral infarction, brain infarction, cerebral ischemia, brain ischemia, upper limb, upper extremity, arm, hand, shoulder, elbow, forearm, finger, wrist, robot*, robot-assisted, exoskeleton, randomized controlled trial, and trial. We will search electronic databases to identify published studies, including CINAHL,

PubMed, Web of Science, MEDLINE, Embase, CNKI, and Wanfang. The draft search strategy is showed in S1 File. We will identify studies published between January 1, 2013, and December 31, 2023. The reference lists of the included studies and relevant reviews will be scanned to identify additional eligible studies.

## Study management and selection

Duplicate studies will be removed by using NoteExpress software. Two reviewers will independently review the titles and abstracts of the remaining studies. Studies that obviously do not meet the inclusion criteria will be excluded. Two reviewers will independently examine the full-text studies according to the eligibility criteria. If there are disagreements, a third reviewer will join in the evaluation. The processes of study selection are shown in Fig 1.

## Data extraction

Two reviewers will independently extract data according to the predefined standard form that includes authors, publication year, age and sex of patients, sample size, types of disease, types of the other interventions, frequency and duration of intervention, outcomes and

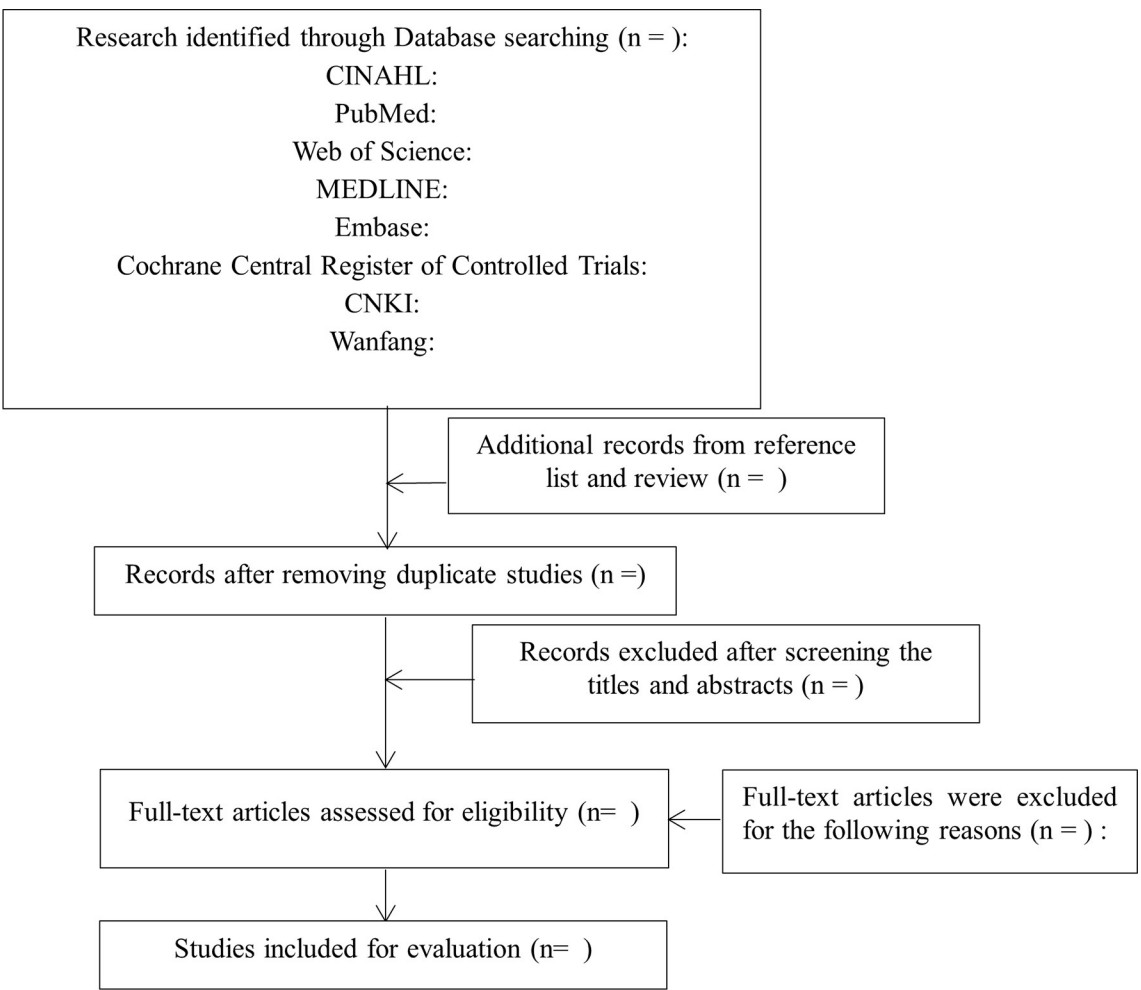

**Fig 1. The processes of study selection.**

measurements (S1 Table). Two reviewers will crosscheck the data extraction, and discrepancies will be resolved through discussions with a third reviewer.

### Risk of bias assessment

Two reviewers will independently assess the methodological quality of all included studies by using the revised version of the Cochrane tool (RoB 2) [17], which has five domains: the randomization process, deviations from intended interventions, missing outcome data, measurement of the outcome, and the selection of the reported result, which will be rated as 'low risk', 'some concerns' or 'high risk'. Two reviewers will discuss the assessment process to compare results, and disagreements will be resolved by discussion with a third reviewer.

### Statistical analysis

We will conduct conventional pairwise meta-analyses to compare all available direct evidence by using STATA version 15.0 software. The pooled effects of continuous data will be calculated by using the standardized mean difference and its 95% confidence interval. Heterogeneity will be assessed by using inconsistency tests with the $I^2$ statistic. If the $P$ value is $\geq 0.1$ and $I^2 \leq$ 50%, we will use the fixed -effects model to synthesize standardized mean difference. Otherwise, we will use the random effects model. Subgroup analysis may be performed to address the problem of high heterogeneity in accordance with the type of disease, country, duration of intervention or age of patients. After the standard paired meta-analysis is completed, we will conduct the network meta-analysis to synthesize all available direct and indirect evidence for each outcome. Due to the potential heterogeneity among the included studies, we will conduct a random-effects network meta-analysis within a Bayesian framework by using Markov Chains Monte Carlo in R version 4.1.3. Each chain will undergo a total of 5000 simulations and posterior summaries will be based on 200 000 subsequent simulations. Network plots will be performed by using STATA version 15.0 software. Based on a loop-special method within each loop of the network [18], the local inconsistency and global inconsistency will be measured in STATA version 15.0 software [19]. Surface under the cumulative ranking curve values will be used to evaluate the ranking of the effectiveness of each intervention [20]. For assessing the publication bias, funnel plots and Egger's regression tests will be performed by using STATA version 15.0 software, if the network meta-analysis model includes more than nine studies [21]. For sensitivity analysis, we will exclude trials judged to be at high risk of bias and assess changes in the effects compared to the primary network meta-analysis and changes in heterogeneity and incoherence of the overall model. We will use the Grading of Recommendations Assessment, Development and Evaluation framework to assess the quality of evidence [22].

## Discussion

Stroke is one of the main causes of long-term disability. Upper limb dysfunction is a common sequela of stroke, and more than 80% of acute stroke patients and more than 40% of chronic stroke patients have this symptom [23]. Upper limb dysfunction can seriously affect the daily life of stroke patients and reduce their quality of life. Consequently, from the perspective of stroke survivors, caregivers and health professionals, the restoration of upper limb function has been identified as a top priority for research [24]. Traditional rehabilitation treatments for the upper limb are time-consuming, laborious and lack therapists, and it is difficult to meet the existing rehabilitation needs. Robot-assisted therapy is an effective measure to improve upper limb function in stroke patients. The main reason is that robot-assisted therapy can provide patients with high-intensity repetitive exercises, which are considered to be the key elements of rehabilitation training [25]. However, some studies have shown that robot-assisted

therapy has the same effect on improving upper limb function as traditional rehabilitation [2, 26]. Combinations of robot-assisted therapy and other rehabilitation programs, such as virtual reality, electromyography, noninvasive brain stimulation and brain-computer interfaces, are considered to be more effective interventions in upper limb rehabilitation [5–7]. To date, there is a lack of research to explore which rehabilitation programs are most effective in combination with robot-assisted therapy.

Therefore, this protocol intends to guide a network meta-analysis to rank the effectiveness of these methods in improving upper limb function in stroke patients. Network meta-analysis is an appropriate method that combines direct and indirect comparisons to analyze the differences in outcomes among multiple interventions. The results of the network meta-analysis will provide a ranking of the effectiveness of each rehabilitation program, which can provide a reference for clinical healthcare professionals in developing rehabilitation plans for stroke patients. Moreover, we also hope that the results of this network meta-analysis can provide some important evidence for recommendations of guidelines.

## Supporting information

**S1 Checklist. PRISMA-P (Preferred Reporting Items for Systematic review and Meta-Analysis Protocols) 2015 checklist: Recommended items to address in a systematic review protocol\*.**
(DOC)

**S1 File. A draft search strategy.**
(DOCX)

**S1 Table. Data extraction form.**
(DOCX)

## Author Contributions

**Conceptualization:** Qian Liu, Fang Wang, Hong Cheng, Xiuying Hu.

**Methodology:** Qian Liu, Li Liu, Xiuying Hu.

**Project administration:** Xiuying Hu.

**Software:** Qian Liu, Yang Xu, Fanyu Zhao.

**Writing – original draft:** Qian Liu.

**Writing – review & editing:** Zuoyan Liu, Yang Xu, Li Liu, Fang Wang, Fanyu Zhao, Hong Cheng, Xiuying Hu.

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
