## [Decision Letter · Decision Letter 0]

7 Feb 2024

PONE-D-23-41381Comparative efficacy of robot-assisted therapy associated with other different interventions on upper limb rehabilitation after stroke: a protocol for a network meta-analysisPLOS ONE

Dear Dr. Hu,

Thank you for submitting your manuscript to PLOS ONE. After careful consideration, we feel that it has merit but does not fully meet PLOS ONE’s publication criteria as it currently stands. Therefore, we invite you to submit a revised version of the manuscript that addresses the points raised during the review process.

Before the article can be procedsed any further, Authors’ must address all Reviewers’ comments. As you will read many concerns have posed, with all of them being relevant in the evaluation process. 

We look forward to receiving your revised manuscript.

Kind regards,

Roberto Di Marco, PhD

Academic Editor

PLOS ONE

Journal Requirements:

Reviewers' comments:

Reviewer's Responses to Questions

**Comments to the Author**

1. Does the manuscript provide a valid rationale for the proposed study, with clearly identified and justified research questions?

Reviewer #1: Partly

Reviewer #2: Partly

2. Is the protocol technically sound and planned in a manner that will lead to a meaningful outcome and allow testing the stated hypotheses?

Reviewer #1: Partly

Reviewer #2: Yes

3. Is the methodology feasible and described in sufficient detail to allow the work to be replicable?

Reviewer #1: No

Reviewer #2: Yes

4. Have the authors described where all data underlying the findings will be made available when the study is complete?

Reviewer #1: No

Reviewer #2: No

5. Is the manuscript presented in an intelligible fashion and written in standard English?

Reviewer #1: No

Reviewer #2: Yes

6. Review Comments to the Author

You may also provide optional suggestions and comments to authors that they might find helpful in planning their study.

Reviewer #1: The protocol focuses on a hot topic and the results may have a significant impact on clinical practice.

I have however some major concerns on clinical and methodological concepts.

From a clinical point of view, the authors neglect the major concept of "opportunity window" for recovery, which would strongly bias results. I suggest either they focus on the first 3/6 months after the event, or conduct a separate analysis for the early/late subacute and chronic phase. Similarly, the the authors pool together very different types of brain lesions, ranging from SH to intracranial hemorrhage and hemorrhagic/ischemic stroke. They need to refocus the etiology to be consistent with the planned title.

Primary outcome measures are heterogeneous. The authors just report a few among the most commonly used scales for body integrity and function (see ICF), but they need to report all the outcome measures (primary and secondary) which they plan to include.

The authors need also to provide a clear list of the associated interventions they plan to include: for example, they briefly mention NIBS. In this case, they should detail which NIBS (rTMS, tDCS, tACS?) and possibly the protocol they will consider. The same holds true for other interventions, with a report of dosing and intensity. This may push up heterogeneity - the authors will need to discuss how they plan to manage this.

From a methodological point of view, the authors do not mention how they will deal with direct, indirect and mixed comparisons, as well as assess inconsistencies.

I would also suggest attaching as supplementary the data extraction form.

A thorough revision by a native speaker is mandatory

Reviewer #2: The manuscript "Comparative efficacy of robot-assisted therapy associated with other different interventions on upper limb rehabilitation after stroke: a protocol for a network meta-analysis" is a protocol-type article with the objective of developing a network meta-analysis, which aims to contribute to determining suitable auxiliary interventions for robot-assisted therapy.

[1. Introduction

The text of the manuscript is very well written, easy to read and understand. The authors contextualize the problem well. However, the authors stated that there is no research comparing which rehabilitation program combined with robot-assisted therapy has the best effect. This is a very strong statement, especially given the large number of research published every day. Therefore, I recommend that authors inform in the introduction how they arrived at this information or improve the text in this section.

[2] Methodology

In the statistical analysis section, the authors need to better justify the techniques to be used, as the selection criteria are not clear.

The authors also did not inform whether they will make the analyzed data available, this is important, as it guarantees the reproducibility of the research.

[3] Discussion

The authors made the following statement: "To date, no studies have explored which rehabilitation programs are most effective in combination with robot-assisted therapy," but they do not say how they reached this conclusion. I recommend that the authors inform how they reached this conclusion or improve the text.

The discussion is redundant, as some parts have already been highlighted very well in the introduction. My recommendation is that the authors discuss the importance of their study and how it can be used, for example, in the formulation of public health policies.

General information

The justification for the research is well defined, but the authors need to make their research questions clearer.

Authors need to describe where all data underlying the results will be made available when the study is completed.

7. PLOS authors have the option to publish the peer review history of their article (what does this mean?). If published, this will include your full peer review and any attached files.

Reviewer #1: No

Reviewer #2: **Yes: **Ricardo Valentim

---

## [Author Response · Author response to Decision Letter 0]

27 Feb 2024

Dear reviewers,

Thank you for your kind advice and providing us the chance to revise our manuscript entitled “Comparative efficacy of robot-assisted therapy associated with other different interventions on upper limb rehabilitation after stroke: a protocol for a network meta-analysis” (ID: PONE-D-23-41381).

Following the suggestion, we have revised the manuscript point by point according to the comments. If we did not get what you mean, please give us another chance to make modifications. The details are as follows.

Responses to reviewer 1:

1. From a clinical point of view, the authors neglect the major concept of “opportunity window”; for recovery, which would strongly bias results. I suggest either they focus on the first 3/6 months after the event, or conduct a separate analysis for the early/late subacute and chronic phase. 

Reply: Thanks for the kind advice. We have reviewed literature and conducted a discussion. We decided that the types of participants focused on the first six months after event and revised the inclusion criteria. (page 5, line 20)

2. The authors pool together very different types of brain lesions, ranging from SH to intracranial hemorrhage and hemorrhagic/ischemic stroke. They need to refocus the etiology to be consistent with the planned title. 

Reply: Thanks for the kind advice. The target disease of this meta-analysis is stroke. Based on our previous search, some studies on intracranial hemorrhage, hemorrhagic/ischemic stroke, cerebral vascular accident and subarachnoid hemorrhage also target stroke. In addition, in researches on robot-assisted rehabilitation, most studies focused on all types of stroke patients, rather than just one type of stroke. Therefore, we considered all types of stroke when formulating the retrieval strategy (page 7, lines 4-7), to avoid obtaining insufficient literature in future literature searches.

3. Primary outcome measures are heterogeneous. The authors just report a few among the most commonly used scales for body integrity and function (see ICF), but they need to report all the outcome measures (primary and secondary) which they plan to include. 

Reply: Thanks for the kind advice. We searched for literature related to this meta-analysis topic and supplemented the outcome measures that used to measure upper limb function and activities of daily living. (page 6, lines 17-22)

4. The authors need also to provide a clear list of the associated interventions they plan to include: for example, they briefly mention NIBS. In this case, they should detail which NIBS (rTMS, tDCS, tACS?) and possibly the protocol they will consider. The same holds true for other interventions, with a report of dosing and intensity. This may push up heterogeneity - the authors will need to discuss how they plan to manage this. 

Reply: Thanks for the kind advice. Based on our understanding, we described the possible interventions that may be considered in types of interventions section (page 6, lines 4-7). Based on clinical work and literature review, the dosage and intensity will be limited to once a day and at least three times a week (page 6, lines 8-9). If we did not get what you mean, please give us another chance to make modifications.

5. From a methodological point of view, the authors do not mention how they will deal with direct, indirect and mixed comparisons, as well as assess inconsistencies. 

Reply: Thanks for the kind advice. We have described the statistical methods in more detail. For direct, indirect and mixed comparisons, as well as assess inconsistencies, we added relevant statistical analysis methods. (page 9, lines 12-22; page 10, lines1-4)

6. I would also suggest attaching as supplementary the data extraction form. 

Reply: Thanks for the kind advice. We attached as supplementary the data extraction form (page 8, line 8).

7. A thorough revision by a native speaker is mandatory

Reply: Thanks for the advice. we invited a scholar, who is a native English speaker and also with technical knowledge of this subject, to assist us during this revision and help us improve the quality of the writing.

Responses to reviewer 2:

1. the authors stated that there is no research comparing which rehabilitation program combined with robot-assisted therapy has the best effect. This is a very strong statement, especially given the large number of research published every day. Therefore, I recommend that authors inform in the introduction how they arrived at this information or improve the text in this section.

Reply: Thanks for the kind advice. After discussion and literature review, we agree that such a statement is indeed inappropriate. Therefore, we revised this sentence in the introduction section. (page 5, lines 7-9)

2. In the statistical analysis section, the authors need to better justify the techniques to be used, as the selection criteria are not clear. 

Reply: Thanks for the kind advice. We have described the statistical methods in more detail. For direct, indirect and mixed comparisons, we added relevant statistical analysis methods. (page 9, lines 12-22; page 10, lines1-4)

3. The authors also did not inform whether they will make the analyzed data available, this is important, as it guarantees the reproducibility of the research.

Reply: Thanks for the advice. Because this meta-analysis will be based on published data and no new data will be generated, we did not inform whether we will make the analyzed data available. The data we will extract and analyze can be obtained through reasonable requests to the corresponding authors. (page 12, lines 2-4)

4. The authors made the following statement: “To date, no studies have explored which rehabilitation programs are most effective in combination with robot-assisted therapy”, but they do not say how they reached this conclusion. I recommend that the authors inform how they reached this conclusion or improve the text. 

Reply: Thanks for the kind advice. After discussion and literature review, we agree that such a statement is indeed inappropriate. Therefore, we revised this sentence in the introduction section. (page 11, lines 8-10)

5. The discussion is redundant, as some parts have already been highlighted very well in the introduction. My recommendation is that the authors discuss the importance of their study and how it can be used, for example, in the formulation of public health policies. 

Reply: Thanks for the kind advice. We added some discussion about the importance of this study. (page 11, lines 11-19)

6. Authors need to describe where all data underlying the results will be made available when the study is completed. 

Reply: Thanks for the advice. We revised the Data Availability section, and when the study is completed, all relevant data underlying the results will be made available through reasonable requests to the corresponding authors. (

---

## [Decision Letter · Decision Letter 1]

10 May 2024

Comparative efficacy of robot-assisted therapy associated with other different interventions on upper limb rehabilitation after stroke: a protocol for a network meta-analysis

PONE-D-23-41381R1

Dear Dr. Hu,

We’re pleased to inform you that your manuscript has been judged scientifically suitable for publication and will be formally accepted for publication once it meets all outstanding technical requirements.

Kind regards,

Roberto Di Marco, PhD

Academic Editor

PLOS ONE

Additional Editor Comments (optional):

Reviewers' comments:

Reviewer's Responses to Questions

**Comments to the Author**

1. Does the manuscript provide a valid rationale for the proposed study, with clearly identified and justified research questions?

Reviewer #2: Yes

2. Is the protocol technically sound and planned in a manner that will lead to a meaningful outcome and allow testing the stated hypotheses?

Reviewer #2: Yes

3. Is the methodology feasible and described in sufficient detail to allow the work to be replicable?

Reviewer #2: Yes

4. Have the authors described where all data underlying the findings will be made available when the study is complete?

Reviewer #2: Yes

5. Is the manuscript presented in an intelligible fashion and written in standard English?

Reviewer #2: Yes

6. Review Comments to the Author

You may also provide optional suggestions and comments to authors that they might find helpful in planning their study.

Reviewer #2: The authors managed to improve the manuscript in relation to the points recommended by the reviewer. Therefore, for my part there is nothing more to recommend.

7. PLOS authors have the option to publish the peer review history of their article (what does this mean?). If published, this will include your full peer review and any attached files.

Reviewer #2: **Yes: **Ricardo Valentim

---

## [Editor Report · Acceptance letter]

20 May 2024

PONE-D-23-41381R1 

PLOS ONE

Dear Dr. Hu, 

I'm pleased to inform you that your manuscript has been deemed suitable for publication in PLOS ONE. Congratulations! Your manuscript is now being handed over to our production team.

Kind regards, 

on behalf of

Dr. Roberto Di Marco 

Academic Editor

PLOS ONE